# Towards Stable and comprehensive Domain Alignment: Max-Margin Domain-Adversarial Training

## Abstract

Domain adaptation tackles the problem of transferring knowledge from a label-rich source domain to an unlabeled or label-scarce target domain. Recently domain-adversarial training (DAT) has shown promising capacity to learn a domain-invariant feature space by reversing the gradient propagation of a domain classifier. However, DAT is still vulnerable in several aspects including (1) training instability due to the overwhelming discriminative ability of the domain classifier in adversarial training, (2) restrictive feature-level alignment, and (3) lack of interpretability or systematic explanation of the learned feature space. In this paper, we propose a novel Max-margin Domain-Adversarial Training (MDAT) by designing an Adversarial Reconstruction Network (ARN). The proposed MDAT stabilizes the gradient reversing in ARN by replacing the domain classifier with a reconstruction network, and in this manner ARN conducts both feature-level and pixel-level domain alignment without involving extra network structures. Furthermore, ARN demonstrates strong robustness to a wide range of hyper-parameters settings, greatly alleviating the task of model selection. Extensive empirical results validate that our approach outperforms other state-of-the-art domain alignment methods. Additionally, the reconstructed target samples are visualized to interpret the domain-invariant feature space which conforms with our intuition.

## 1 Introduction

Deep neural networks have gained great success on a wide range of tasks such as visual recognition and machine translation (LeCun et al., 2015). They usually require a large number of labeled data that can be prohibitively expensive to collect, and even with sufficient supervision their performance can still be poor when being generalized to a new environment. The problem of discrepancy between the training and testing data distribution is commonly referred to as *domain shift* (Shimodaira, 2000). To alleviate the effect of such shift, *domain adaptation* sets out to obtain a model trained in a label-rich source domain to generalize well in an unlabeled target domain. Domain adaptation has benefited various applications in many practical scenarios, including but not limited to object detection under challenging conditions (Chen et al., 2018), cost-effective learning using only synthetic data to generalize to real-world imagery (Vazquez et al., 2013), etc.

Prevailing methods for unsupervised domain adaptation (UDA) are mostly based on *domain alignment* which aims to learn domain-invariant features by reducing the distribution discrepancy between the source and target domain using some pre-defined metrics such as maximum mean discrepancy (Tzeng et al., 2014). Recently, Ganin & Lempitsky (2015) proposed to achieve domain alignment by domain-adversarial training (DAT) that reverses the gradients of a domain classifier to maximize domain confusion. Having yielded remarkable performance gain, DAT was employed in many subsequent UDA methods (Long et al., 2018; Shu et al., 2018). Even so, there still exist three critical issues of DAT that hinder its performance: (1) as the domain classifier has high-capacity to discriminate two domains, the unbalanced adversarial training cannot continuously provide effective gradients, which is usually overcome by manually adjusting the weights of adversarial training according to specific tasks; (2) DAT-based methods cannot deal with pixel-level *domain shift* (Hoffman et al., 2018); (3) the domain-invariant features learned by DAT are only based on intuition but difficult to interpret, which impedes the investigation of the underlying mechanism of adversarial domain adaptation.

To overcome the aforementioned difficulties, we propose an innovative DAT approach, namely Max-margin Domain-Adversarial Training (MDAT), to realize stable and comprehensive domain alignment. To demonstrate its effectiveness, we develop an Adversarial Reconstruction Network (ARN) that only utilizes MDAT for UDA. Specifically, ARN consists of a shared feature extractor, a label predictor, and a reconstruction network (i.e. decoder) that serves as a domain classifier. Supervised learning is conducted on source domain, and MDAT helps learn domain-invariant features. In MDAT, the decoder only focuses on reconstructing samples on source domain and pushing the target domain away from a margin, while the feature extractor aims to fool the decoder by learning to reconstruct samples on target domain. In this way, three critical issues can be solved by MDAT: (1) the max-margin loss reduces the discriminative capacity of domain classifier, leading to balanced and thus stable adversarial training; (2) without involving new network structures, MDAT achieves both pixel-level and feature-level domain alignment; (3) visualizing the reconstructed samples reveals how the source and target domains are aligned. We evaluate ARN with MDAT on five visual and non-visual UDA benchmarks. It achieves significant improvement to DAT on all tasks with pixel-level or higher-level *domain shift*. We also observe that it is insensitive to the choices of hyperparameters and as such is favorable for replication in practice. In principle, our approach is generic and can be used to enhance any UDA methods that leverage domain alignment as an ingredient.

## 2 Related Work

Domain adaptation aims to transfer knowledge from one domain to another. Ben-David et al. (2010) provide an upper bound of the test error on the target domain in terms of the source error and the $\mathcal{H}\triangle\mathcal{H}$-distance. As the source error is stationary for a fixed model, the goal of most UDA methods is to minimize the $\mathcal{H}\triangle\mathcal{H}$-distance by reducing some metrics such as Maximum Mean Discrepancy (MMD) (Tzeng et al., 2014; Long et al., 2015) and CORAL (Sun & Saenko, 2016). Inspired by Generative Adversarial Networks (GAN) (Goodfellow et al., 2014), Ganin & Lempitsky (2015) proposed to learn domain-invariant features by adversarial training, which has inspired many UDA methods thereafter. Adversarial Discriminative Domain Adaptation (ADDA) tried to fool the label classifier by adversarial training but not in an end-to-end manner. CyCADA (Hoffman et al., 2018) and PixelDA (Bousmalis et al., 2017) leveraged GAN to conduct both feature-level and pixel-level domain adaptation, which yields significant improvement yet the network complexity is high.

Another line of approaches that are relevant to our method is the reconstruction network (i.e. the decoder network). The success of image-to-image translation corroborates that it helps learn pixel-level features in an unsupervised manner. In UDA, Ghifary et al. (2016) employed a decoder network for pixel-level adaptation, and Domain Separate Network (DSN) (Bousmalis et al., 2016) further leveraged multiple reconstruction networks to learn domain-specific features. These approaches treat the decoder network as an independent component that is irrelevant to domain alignment (Glorot et al., 2011). In this paper, our approach proposes to utilize the decoder network as domain classifier in MDAT which enables both feature-level and pixel-level domain alignment in a stable and straightforward fashion.

## 3 Problem Formulation

### 3.1 Problem Definition and Notations

In unsupervised domain adaptation, we assume that the model works with a labeled dataset $\mathbf{X}_S$ and an unlabeled dataset $\mathbf{X}_T$. Let $\mathbf{X}_S = \{(\mathbf{x}_i^s, y_i^s)\}_{i \in [N_s]}$ denote the labeled dataset of $N_s$ samples from the source domain, and the certain label $y_i^s$ belongs to the label space $Y$ that is a finite set ($Y = 1, 2, ..., K$). The other dataset $\mathbf{X}_T = \{\mathbf{x}_i^t\}_{i \in [N_t]}$ has $N_t$ samples from the target domain but has no labels. We further assume that two domains have different distributions, i.e. $\mathbf{x}_i^s \sim \mathcal{D}_S$ and $\mathbf{x}_i^t \sim \mathcal{D}_T$. In other words, there exist some *domain shift* (Ben-David et al., 2010) between $\mathcal{D}_S$ and $\mathcal{D}_T$. The ultimate goal is to learn a model that can predict the label $y_i^t$ given the target input $\mathbf{x}_i^t$.

### 3.2 Imbalanced Minimax Game in Domain-Adversarial Training

To achieve domain alignment, Domain-Adversarial Training (DAT) is a minimax game between a shared feature extractor $F$ for two domains and a domain classifier $D$. The domain classifier is

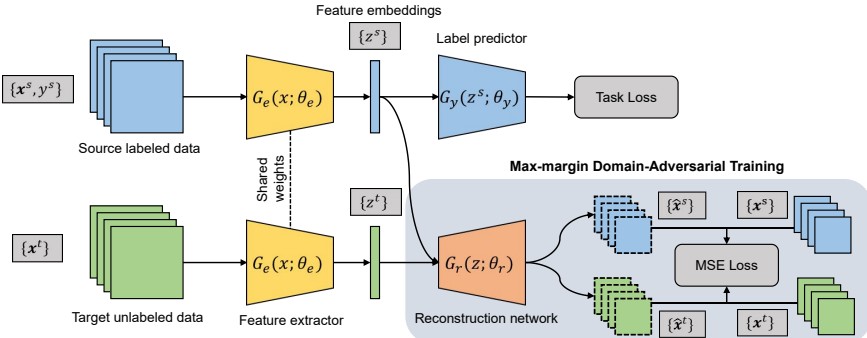

Figure 1: The proposed architecture is composed of a shared feature extractor $G_e$ for two domains, a label predictor $G_y$ and a reconstruction network $G_r$. In addition to the basic supervised learning in the source domain, our adversarial reconstruction training enables the extractor $G_e$ to learn domain-invariant features. Specifically, the network $G_r$ aims to reconstruct the source samples $x^s$ and to impede the reconstruction of the target samples $x^t$, while the extractor $G_e$ tries to fool the reconstruction network in order to reconstruct the target samples $x^t$.

trained to determine whether the input sample belongs to the source or the target domain while the feature extractor learns to deceive the domain classifier, which is formulated as:

$$\min_F \max_D \mathcal{L}_{DAT}(D_s, D_t) = \mathbb{E}_{x \sim D_s}[\ln F(x)] + \mathbb{E}_{x \sim D_t}[\ln (1 - D(F(x)))]. \tag{1}$$

In DAT, we usually utilize CNN as the feature extractor and fully connected layers (FC) as the domain classifier. DAT reduces the cross-domain discrepancy, achieving significant performance improvement for UDA. Nevertheless, the training of DAT is rather unstable. Without sophisticated tuning of the hyper-parameters, DAT cannot reach the convergence. Through empirical experiments, we observe that such instability is due to the imbalanced minimax game. The binary domain classifier $D$ can easily achieve convergence with very high accuracy at an early training epoch, while it is much harder for the feature extractor $F$ to fool the domain classifier and to simultaneously perform well on the source domain. In this sense, the domain classifier dominates DAT, and the only solution is to palliate the training of $D$ by tuning the hyper-parameters according to different tasks. In our method, we restrict the capacity of the domain classifier so as to form a minimax game in a harmonious manner. Inspired by the max-margin loss in Support Vector Machine (SVM) (Cristianini et al., 2000) (i.e. hinge loss), if we push the source domain and the target domain away from a margin rather than as far as possible, then the training task of $F$ to fool $D$ becomes easier. For a binary domain classifier, we define the margin loss as

$$\mathcal{L}_{margin}(y) = [0, m - t \cdot y]^+, \tag{2}$$

where $y$ is the predicted domain label, $[\cdot]^+ := max(0, \cdot)$, $m$ is a positive margin and $t$ is the ground truth label for two domains ($t = -1$ for the source domain and $t = 1$ for the target domain). Then we introduce our MDAT scheme based on an innovative network architecture.

### 3.3 MAX-MARGIN DOMAIN-ADVERSARIAL TRAINING

Besides the training instability issue, DAT also suffers from restrictive feature-level alignment – lack of pixel-level alignment. To realize stable and comprehensive domain alignment together, we first propose an Adversarial Reconstruction Network (ARN) and then elaborate MDAT.

As depicted in Figure 1, our model consists of three parts including a shared feature extractor $G_e$ for both domains, a label predictor $G_y$ and a reconstruction network $G_r$. Let the feature extractor $G_e(\mathbf{x}; \theta_e)$ be a function parameterized by $\theta_e$ which maps an input sample $\mathbf{x}$ to a deep embedding $\mathbf{z}$. Let the label predictor $G_y(\mathbf{z}; \theta_y)$ be a task-specific function parameterized by $\theta_y$ which maps an embedding $\mathbf{z}$ to a task-specific prediction $\hat{y}$. The reconstruction network $G_r(\mathbf{z}; \theta_r)$ is a decoding function parameterized by $\theta_r$ that maps an embedding $\mathbf{z}$ to its corresponding reconstruction $\hat{\mathbf{x}}$.

The first learning objective for the feature extractor $G_e$ and label predictor $G_y$ is to perform well in the source domain. For a supervised K-way classification problem, it is simply achieved by minimizing

the negative log-likelihood of the ground truth class for each sample:

$$\mathcal{L}_{task} = \sum_{i=1}^{N_s} \mathcal{L}_y(\mathbf{x}_i^s, \mathbf{y}_i^s) = -\sum_{i=1}^{N_s} \mathbf{y}_i^s \cdot \log G_y(G_e(\mathbf{x}_i^s; \theta_e); \theta_y), \tag{3}$$

where $\mathbf{y}_i^s$ is the one-hot encoding of the class label $y_i^s$ and the logarithm operation is conducted on the softmax predictions of the model.

The second objective is to render the feature learning to be domain-invariant. This is motivated by the *covariate shift* assumption (Shimodaira, 2000) that indicates if the feature distributions $S(\mathbf{z}) = \{G_e(\mathbf{x}; \theta_e) | \mathbf{x} \sim \mathcal{D}_S\}$ and $T(\mathbf{z}) = \{G_e(\mathbf{x}; \theta_e) | \mathbf{x} \sim \mathcal{D}_T\}$ are similar, the source label predictor $G_y$ can achieve a similar high accuracy in the target domain. To this end, we design a decoder network $G_r$ that serves as a domain classifier, and then MDAT could be applied for stable training. Different from the normal binary domain classifier, MDAT lets the decoder network $G_r$ only reconstruct the features in the source domain and push the features in the target domain away from a margin $m$. In this way, the decoder has the functionality of distinguishing the source domain from the target domain. The objective of training $G_r$ is formulated as

$$\min_{\theta_r} \sum_{i=1}^{N_s+N_t} \mathcal{L}_{margin}(\mathcal{L}_r(\mathbf{x}_i)) = \min_{\theta_r} \sum_{i=1}^{N_s} \mathcal{L}_r(\mathbf{x}_i^s) + \sum_{j=1}^{N_t} [m - \mathcal{L}_r(\mathbf{x}_j^t)]^+, \tag{4}$$

where $m$ is a positive margin and $\mathcal{L}_r(\cdot)$ is the mean squared error (MSE) term for the reconstruction loss that is defined as

$$\mathcal{L}_r(\mathbf{x}) = ||G_r(G_e(\mathbf{x}; \theta_e); \theta_r) - \mathbf{x}||_2^2, \tag{5}$$

where $|| \cdot ||_2^2$ denotes the squared $L_2$-norm.

Oppositely, to form a minimax game, the feature extractor $G_e$ learns to deceive $G_r$ such that the learned target features are indistinguishable to the source ones, which is formulated by:

$$\min_{\theta_e} \sum_{j=1}^{N_t} \mathcal{L}_r(\mathbf{x}_j^t). \tag{6}$$

Then the whole learning procedure of ARN with MDAT can be formulated by:

$$\min_{\theta_e, \theta_c} \sum_{i=1}^{N_s} \mathcal{L}_y(\mathbf{x}_i^s, \mathbf{y}_i^s) + \alpha \sum_{j=1}^{N_t} \mathcal{L}_r(\mathbf{x}_j^t), \tag{7}$$

$$\min_{\theta_r} \sum_{i=1}^{N_s} \mathcal{L}_r(\mathbf{x}_i^s) + \sum_{j=1}^{N_t} [m - \mathcal{L}_r(\mathbf{x}_j^t)]^+, \tag{8}$$

where $\mathcal{L}_y$ denotes the negative log-likelihood of the ground truth class for labeled sample $(\mathbf{x}_i^s, \mathbf{y}_i^s)$ and $\alpha$ controls the interaction of the loss terms. In the following section, we provide theoretical justifications on how MDAT reduces the distribution discrepancy, and discuss why it is superior to the classic DAT.

### 3.4 THEORETICAL JUSTIFICATIONS

In this section, we provide the theoretical justifications on how the proposed method reduces the distribution discrepancy for UDA. The rationale behind domain alignment is motivated from the learning theory of non-conservative domain adaptation problem by Ben-David et al. (Ben-David et al., 2010):

**Theorem 3.1** *Let $\mathcal{H}$ be the hypothesis space where $h \in \mathcal{H}$. Let $(\mathcal{D}_S, \epsilon_s)$ and $(\mathcal{D}_T, \epsilon_t)$ be the two domains and their corresponding generalization error functions. The expected error for the target domain is upper bounded by*

$$\epsilon_t(h) \leq \epsilon_s(h) + \frac{1}{2} d_{\mathcal{H} \triangle \mathcal{H}}(\mathcal{D}_S, \mathcal{D}_T) + \lambda, \forall h \in \mathcal{H}, \tag{9}$$

*where $d_{\mathcal{H} \triangle \mathcal{H}}(\mathcal{D}_S, \mathcal{D}_T) = 2 \sup_{h_1, h_2 \in \mathcal{H}} |\Pr_{x \sim \mathcal{D}_S}[h_1(x) \neq h_2(x)] - \Pr_{x \sim \mathcal{D}_T}[h_1(x) \neq h_2(x)]|$ and $\lambda = \min_h [\epsilon_s(h) + \epsilon_t(h)]$.*

Theoretically, when we minimize the $\mathcal{H}\triangle\mathcal{H}$-distance, the upper bound of the expected error for the target domain is reduced accordingly. As derived in DAT (Ganin & Lempitsky, 2015), assuming a family of domain classifiers $\mathcal{H}_d$ to be rich enough to contain the symmetric difference hypothesis set of $\mathcal{H}_p$, such that $\mathcal{H}_p\triangle\mathcal{H}_p = \{h|h = h_1 \oplus h_2, h_1, h_2 \in \mathcal{H}_p\}$ where $\oplus$ is XOR-function, the empirical $\mathcal{H}_p\triangle\mathcal{H}_p$-distance has an upper bound with regard to the optimal domain classifier $h$:

$$d_{\mathcal{H}_p\triangle\mathcal{H}_p}(\hat{\mathcal{D}}_S, \hat{\mathcal{D}}_T) \le 2 \sup_{h\in\mathcal{H}_d} | \Pr_{\mathbf{z}\sim\hat{\mathcal{D}}_S} [h(\mathbf{z}) = 0] + \Pr_{\mathbf{z}\sim\hat{\mathcal{D}}_T} [h(\mathbf{z}) = 1] - 1|, \tag{10}$$

where $\hat{\mathcal{D}}_S$ and $\hat{\mathcal{D}}_T$ denote the distributions of the source and target feature space $\mathcal{Z}_S$ and $\mathcal{Z}_T$, respectively. Note that the MSE of $G_r$ plus a ceiling function is a form of domain classifier $h(\mathbf{z})$, i.e. $\lceil[m - \mathcal{L}_r(\cdot)]^+ - 0.5\rceil$ for $m = 1$. It maps source samples to 0 and target samples to 1 which is exactly the upper bound in Eq.10. Therefore, our reconstruction network $G_r$ maximizes the domain discrepancy with a margin and the feature extractor learns to minimize it oppositely.

### 3.5 DISCUSSIONS

Compared with the conventional DAT-based methods that are usually based on a binary logistic network (Ganin & Lempitsky, 2015), the proposed ARN with MDAT is more attractive and incorporates new merits conceptually and theoretically:

**(1) Stable training and insensitivity to hyper-parameters.** Using the decoder as domain classifier with a margin loss to restrain its overwhelming capacity in adversarial training, the minimax game can continuously provide effective gradients for training the feature extractor. Moreover, through the experiments in Section 4, we discover that our method shows strong robustness to the hyper-parameters, i.e. $\alpha$ and $m$, greatly alleviating the parameters tuning for model selection.

**(2) Richer information for comprehensive domain alignment.** Rather than DAT that uses a bit of domain information, MDAT utilizes the reconstruction network as the domain classifier that could capture more domain-specific and pixel-level features during the unsupervised reconstruction (Bousmalis et al., 2016). Therefore, MDAT further helps address pixel-level domain shift apart from the feature-level shift, leading to comprehensive domain alignment in a straightforward manner.

**(3) Feature visualization for method validation.** Another key merit of MDAT is that MDAT allows us to visualize the features directly by the reconstruction network. It is crucial to understand to what extent the features are aligned since this helps to reveal the underlying mechanism of adversarial domain adaptation. We will detail the interpretability of these adapted features in Section 4.3.

## 4 EXPERIMENT

In this section, we evaluate the proposed ARN with MDAT on a number of visual and non-visual UDA tasks with varying degrees of *domain shift*. We conduct ablation study to corroborate the effectiveness of MDAT and unsupervised reconstruction for UDA. Then the sensitivity of the hyperparameters is investigated, and the adapted features are interpreted via the reconstruction network in ARN.

**Setup.** We evaluate our method on four classic visual UDA datasets and a WiFi-based Gesture Recognition (WGR) dataset (Zou et al., 2019). The classic datasets have middle level of domain shift including MNIST (LeCun et al., 1998), USPS (Hull, 1994), Street View House Numbers (SVHN) (Netzer et al., 2011) and Synthetic Digits (SYN). For a fair comparison, we follow the same CNN architecture as DANN (Ganin & Lempitsky, 2015) while using the inverse of $G_e$ as $G_r$ with pooling operation replaced by upsampling. For the penalty term $\alpha$, we choose 0.02 by searching over the grid $\{10^{-2}, 1\}$. We also obtain the optimal margin $m = 5$ by a search over $\{10^{-1}, 10\}$. Then we use the **same hyperparameter settings** for all tasks to show the robustness. For the optimization, we simply use Adam Optimizer ($lr = 2 \times 10^{-4}, \beta_1 = 0.5, \beta_2 = 0.999$) and train all experiments for 50 epochs with batch size 128. We implemented our model and conducted all the experiments using the **PyTorch** framework. More implementation details are illustrated in the appendix.

**Baselines.** We evaluate the efficacy of our approach by comparing it with existing UDA methods that perform three ways of domain alignment. Specifically, MMD regularization (Long et al., 2015) and Correlation Alignment (Sun & Saenko, 2016) employ the statistical distribution matching. DRCN (Ghifary et al., 2016) and DSN (Bousmalis et al., 2016) use the reconstruction error for UDA,

| | Source
Target | MNIST
USPS | USPS
MNIST | SVHN
MNIST | SYN
SVHN |
|---|---|---|---|---|---|
| *Source-Only model* | | 78.2 | 63.4 | 54.9 | 86.7 |
| *Train on target* | | 96.5 | 99.4 | 99.4 | 91.3 |
| [S] MMD (Long et al., 2015) | | 81.1 | - | 71.1 | 88.0 |
| [S] CORAL (Sun & Saenko, 2016) | | 80.7 | - | 63.1 | 85.2 |
| [R] DRCN* (Ghifary et al., 2016) | | 91.8 | 73.7 | 82.0 | 87.5 |
| [R] DSN (Bousmalis et al., 2016) | | 91.3 | - | 82.7 | 91.2 |
| [A] DANN (Ganin et al., 2016) | | 85.1 | 73.0 | 74.7 | 90.3 |
| [A] ADDA (Tzeng et al., 2017) | | 89.4 | 90.1 | 76.0 | - |
| [A] CyCADA (Hoffman et al., 2018) | | 95.6 | 96.5 | 90.4 | - |
| [A] CADA (Zou et al., 2019) | | 96.4 | 97.0 | 90.9 | - |
| [A] MECA (Morerio et al., 2018) | | - | - | 95.2 | 90.3 |
| **ARN w.o. MDAT** | | 93.1±0.3 | 76.5±1.2 | 67.4±0.9 | 86.8±0.5 |
| **ARN with MDAT (proposed)** | | **98.6±0.3** | **98.4±0.1** | **97.4±0.3** | **92.0±0.2** |

Table 1: We compare with general, statistics-based (**S**), reconstruction-based (**R**) and adversarial-based (**A**) state-of-the-art approaches. We repeated each experiment for 3 times and report the average and standard deviation (std) of the test accuracy in the target domain.

while many prevailing UDA methods adopt domain-adversarial training including DANN (Ganin & Lempitsky, 2015), ADDA (Tzeng et al., 2017), MECA (Morerio et al., 2018), CyCADA (Hoffman et al., 2018) and CADA (Zou et al., 2019). For all transfer tasks, we follow the same protocol as DANN (Ganin & Lempitsky, 2015) that uses official training data split in both domains for training and evaluates the testing data split in the target domain.

## 4.1 Overall Results

**MNIST↔USPS.** Both datasets are composed of grey-scale handwritten images with diverse stroke weights, leading to low-level *domain shift*. Since USPS has only 7291 training images, **USPS→MNIST** is more difficult. As shown in Table 1, our method achieves state-of-the-art accuracy of 98.6% on **MNIST→USPS** and 98.4% on **USPS→MNIST**, which demonstrates that ARN can tackle low-level *domain shift* by only using ART (rather than many adversarial UDA methods that adopt other loss terms to adjust classifier boundaries or conduct style transfer).

**SVHN→MNIST and SYN→SVHN.** The SVHN dataset contains RGB digit images that introduce significant variations such as scale, background, embossing, rotation, slanting and even multiple digits. The SYN data consists of 50$k$ RGB images of varying color, background, blur and orientation. These two tasks have tremendous pixel-level domain shfit. The proposed method achieves a state-of-the-art performance of 97.4% for SVHN→MNIST, far ahead of other DAT-based methods, significantly improving the classic DANN by 22.7%. Similarly, ARN with MDAT also achieves a noticeable improvement of 5.3% compared with the source-only model, even outperforming the supervised SVHN accuracy 91.3%.

Table 2: Comparisons on WGR.

| Source
Target | Room A
Room B |
|---|---|
| Method | Accuracy (%) |
| *Source-only* | 58.4±0.7 |
| [S] MMD | 61.2±0.5 |
| [R] DRCN | 69.3±0.3 |
| [A] DANN | 68.2±0.2 |
| [A] ADDA | 71.5±0.3 |
| [A] CADA | 88.8±0.1 |
| **ARN+MDAT** | **91.3±0.2** |

**WiFi Gesture Recognition with Distant Domains.** To evaluate the proposed method on a non-visual UDA task, we applied our method to the WiFi gesture recognition dataset (Zou et al., 2019). The WiFi data of six gestures was collected in two rooms regarded as two domains. The results in Table 2 demonstrate that our approach significantly improves classification accuracy against Source-Only and DANN by 32.9% and 23.1%, respectively.

| $\alpha$ | 0.01 | 0.03 | 0.07 | 0.1 | 0.2 | 0.3 | 0.5 | 1.0 |
|---|---|---|---|---|---|---|---|---|
| DANN | 71.1 | 74.1 | 72.7 | 74.1 | 74.7 | 9.6 | 9.7 | 10.3 |
| ARN ($m = 1$) | 95.7 | 95.9 | 93.3 | 93.2 | 80.1 | 75.3 | 73.1 | 67.5 |

| $m$ | 0.1 | 0.3 | 0.5 | 0.7 | 1.0 | 2.0 | 5.0 | 10.0 |
|---|---|---|---|---|---|---|---|---|
| ARN ($\alpha = 2e^{-2}$) | 64.5 | 75.2 | 90.0 | 92.6 | 96.0 | 97.4 | 97.7 | 96.7 |

Table 3: The accuracy (%) with different hyperparameters on **SVHN→MNIST**.

## 4.2 ABLATION STUDY AND SENSITIVITY ANALYSIS

**The contribution of MDAT and image reconstruction in ARN.** We design an ablation study to verify the contribution of MDAT and unsupervised reconstruction in ARN. To this end, we discard the term $\mathcal{L}_r(\mathbf{x}^t)$ in Eq.4, and evaluate the method, denoted as ARN w.o. MDAT in Table 1. (1) Comparing ARN w.o. MDAT with source-only model, we can infer the effect of unsupervised reconstruction for UDA. It is observed that ARN w.o. MDAT improves tasks with low-level *domain shift* such as **MNIST↔USPS**, which conforms with our discussion that the unsupervised reconstruction is instrumental in learning low-level features. (2) Comparing ARN w.o. MDAT with the original ARN, we can infer the contribution of MDAT. Table 1 shows that the MDAT achieves an impressive margin-of-improvement. For **USPS→MNIST** and **SVHN→MNIST**, the MDAT improves ARN w.o. MDAT by around 30%. It demonstrates that MDAT which helps learn domain-invariant representations is the main reason for the tremendous improvement.

**Parameter sensitivity.** We investigate the effect of $\alpha$ and $m$ on **SVHN→MNIST**. The results in Table 3 show that ARN achieves good performance as $\alpha \in [0.01, 0.1]$ and even with larger $\alpha$ ARN is able to achieve convergence. In comparison, denoting $\alpha$ as the weight of adversarial loss, the DANN cannot converge when $\alpha > 0.2$. For the sensitivity of $m$, the accuracy of ARN exceeds 96.0% as $m \geq 1$. These analyses validate that the training of ARN is not sensitive to the parameters and even in the worst cases ARN can achieve convergence.

**Gradients and training procedure.** We draw the training procedure with regard to loss and target accuracy in Figure 2(b) and Figure 2(a), respectively. In Figure 2(b), ARN has smoother and more effective gradients ($\mathcal{L}_r$) for all $\alpha$, while the loss of DAT domain classifier ($\mathcal{L}_d$) gets extremely small at the beginning. This observation conforms with our intuition, which demonstrates that by restricting the capacity of domain classifier MDAT provides more effective gradients for training feature extractor, leading to a more stable training procedure. This could be further validated in Figure 2(b) where the ARN accuracy is more stable than that of DAT across training epochs.

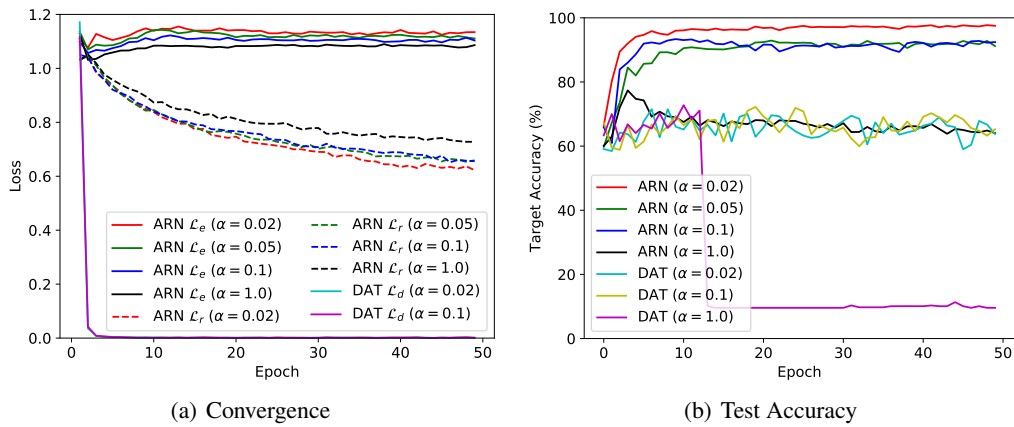

(a) Convergence          (b) Test Accuracy

Figure 2: The training procedure with regard to loss and test accuracy. ($\mathcal{L}_e :=$ Eq. 6; $\mathcal{L}_r :=$ Eq. 4; $\mathcal{L}_d$ is the domain loss of DAT (Ganin & Lempitsky, 2015); $\alpha$ is the penalty term of $\mathcal{L}_e$ and $\mathcal{L}_d$.)

| | Source Images | Target Images | R-Target Images |
|---|---|---|---|
| **MNIST→USPS** | | | |
| **USPS→MNIST** | | | |
| **SVHN→MNIST** | | | |
| **SYN→SVHN** | | | |

Table 4: Visualizing the source image, target images and reconstructed target images (R-Target Images) for four digit adaptation tasks.

### 4.3 VISUALIZATION AND ANALYSIS

**Interpreting MDAT features via reconstructed images.** One of the key advantages of ARN is that by visualizing the reconstructed target images we can infer how the features are domain-invariant. We reconstruct the MDAT features of the test data and visualize them in Table 4. It is observed that the target features are reconstructed to source-like images by the decoder $G_r$. As discussed before, intuitively, MDAT forces the target features to mimic the source features, which conforms with our visualization. Similar to image-to-image translation, this indicates that our method conducts implicit feature-to-feature translation that transfers the target features to source-like features, and hence the features become domain-invariant.

**T-SNE embeddings.** We analyze the performance of domain alignment for DANN (DAT) (Ganin & Lempitsky, 2015) and ARN (MDAT) by plotting T-SNE embeddings of the features **z** on the task **SVHN→MNIST**. In Figure 3(a), the source-only model obtains diverse embeddings for each category but the domains are not aligned. In Figure 3(b), the DANN aligns two domains but the decision boundaries of the classifier are vague. In Figure 3(c), the proposed ARN effectively aligns two domains for all categories and the classifier boundaries are much clearer.

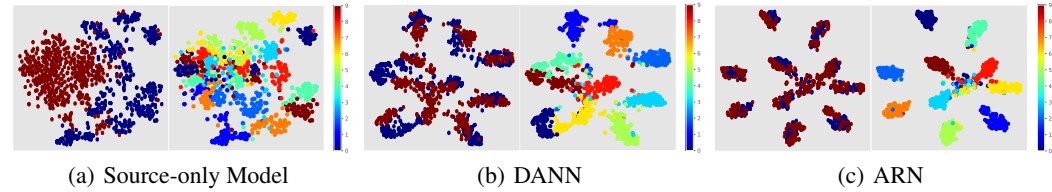

(a) Source-only Model          (b) DANN          (c) ARN

Figure 3: T-SNE visualization on **SVHN→MNIST** with their corresponding domain labels (red: target; blue: source) and category labels (10 classes) shown in the left and right subfigures, respectively.

## 5 CONCLUSION

We proposed a new domain alignment approach namely max-margin domain-adversarial training (MDAT) and a MDAT-based network for unsupervised domain adaptation. The proposed method offers effective and stable gradients for the feature learning via an adversarial game between the feature extractor and the reconstruction network. The theoretical analysis provides justifications on how it minimizes the distribution discrepancy. Extensive experiments demonstrate the effectiveness of our method and we further interpret the features by visualization that conforms with our insight. Potential evaluation on semi-supervised learning constitutes our future work.

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

APPENDIX

IMPLEMENTATION DETAILS

**Hyperparameter** For all tasks, we simply use the same hyperparameters that are chosen from the sensitivity analysis. We use $\alpha = 0.02$ and $m = 5.0$, and we reckon that better results can be obtained by tuning the hyperparameters for specific tasks.

**Network Architecture** For a fair comparison, we follow the network in DANN (Ganin & Lempitsky, 2015) for digit adaptation and simply build the reconstruction network by the inverse network of the extractor. Here we draw the network architectures in Table 5. For WiFi gesture recognition, we adopt the same architecture as CADA (Zou et al., 2019) that is a modified version of LeNet-5.

| Layer Index | Feature Extractor | Decoder Network | Label Predictor |
|---|---|---|---|
| 0 | $32 \times 32 \times 3$ Image | | |
| 1 | $5 \times 5$ conv. 64 ReLU | 2048 dense, ReLU | 10 dense, softmax |
| 2 | $3 \times 3$ max-pool, stride 2 | 3072 dense, ReLU | |
| 3 | $5 \times 5$ conv. 64 ReLU | $5 \times 5$ conv. 128 ReLU | |
| 4 | $3 \times 3$ max-pool, stride 2 | upsample 2 | |
| 5 | $5 \times 5$ conv. 128 ReLU | $5 \times 5$ conv. 64 ReLU | |
| 6 | 3072 dense, dropout, ReLU | upsample 2 | |
| 7 | 2048 dense, dropout ReLU | $5 \times 5$ conv. 64 ReLU | |

Table 5: The network architecture used in the experiments.

SENSITIVITY

We have presented all the results of the sensitivity study in Section 4.2, and now we show their detailed training procedures in Figure 4(a) and 4(b). It is observed that the accuracy increases when $\alpha$ drops or the margin $m$ increases. The reason is very simple: (1) when $\alpha$ is too large, it affects the effect of supervised training on source domain; (2) when the margin $m$ is small, the divergence between source and target domain (i.e. $\mathcal{H}\triangle\mathcal{H}$-distance) cannot be measured well.

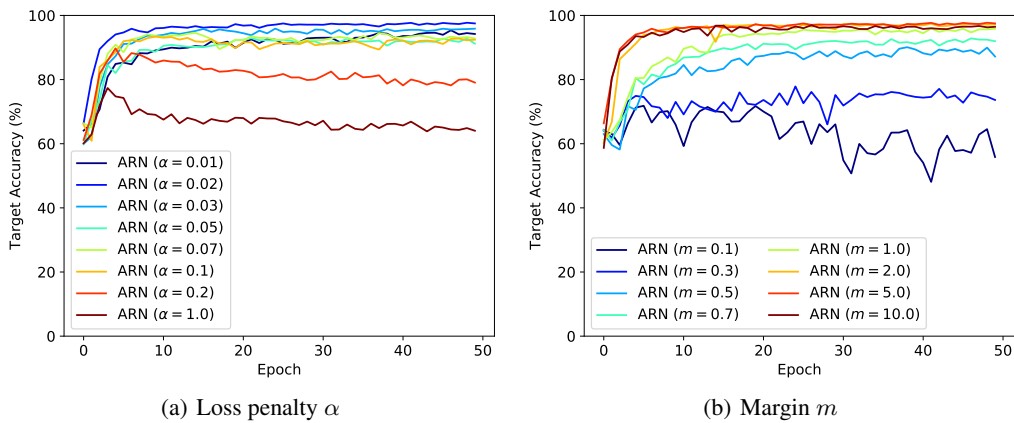

(a) Loss penalty $\alpha$          (b) Margin $m$

Figure 4: The training procedure of ARN with different hyper-parameters.

VISUALIZATION

Here we provide more visualization of the reconstructed images of target samples. In Figure 5, the target samples are shown in the left column while their corresponding reconstructed samples are shown in the right. We can see that for low-level *domain shift* such as **MNIST↔USPS**, the reconstructed target samples are very source-like while preserving their original shapes and skeletons. However, for larger *domain shift* in Figure 5(c) and 5(d), they are reconstructed to source-like same digits but simultaneously some noises are removed. Specifically, in Figure 5(d), we can see that one target sample (SVHN) may contain more than one digits that are noises for recognition. After reconstruction, only the right digits are reconstructed. Some target samples may suffer from terrible illumination conditions but their reconstructed digits are very clear, which is amazing.

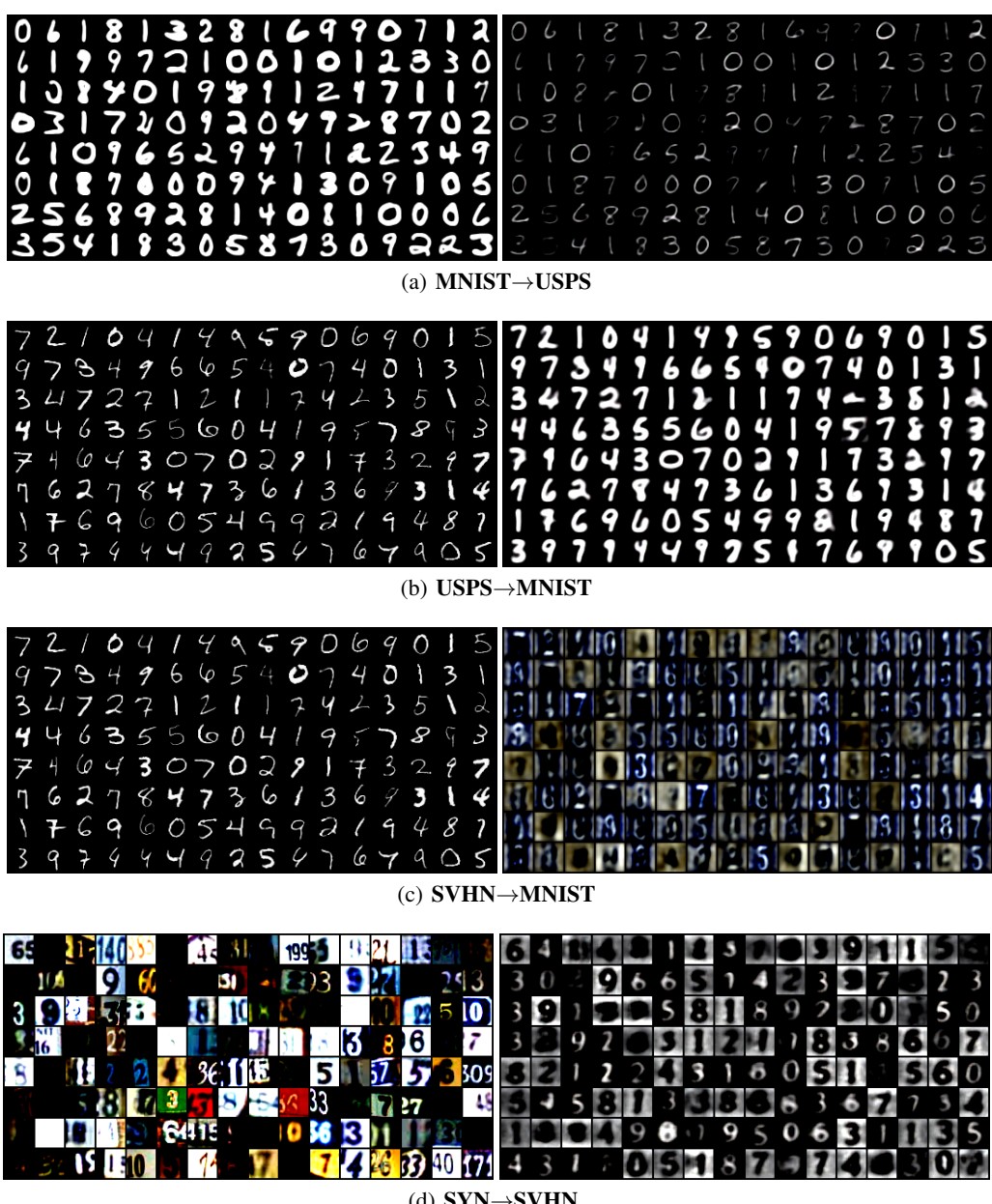

(a) **MNIST→USPS**

(b) **USPS→MNIST**

(c) **SVHN→MNIST**

(d) **SYN→SVHN**

Figure 5: Visualization of the target samples and their corresponding reconstructed target samples.

