# OpenReview forum: "Towards Stable and comprehensive Domain Alignment: Max-Margin Domain-Adversarial Training"
_ICLR.cc/2020/Conference — Reject_

### Official Review · AnonReviewer3 · 2019-10-16
**Official Blind Review #3**

**Rating:** 3

**Review:**

This paper proposed a new method for unsupervised domain adaptation. Different from a conventional domain classifier based adaptation, they propose to utilize the loss of autoencoder to extract domain-invariant features. They trained reconstruction network to reconstruct source examples well whereas making reconstruction loss of the target examples large with some margin. Their goal is to stabilize the training of adversarial training for domain adaptation, incorporate pixel-level information, and give interpretable learned feature space. They performed experiments on digits datasets and WiFi Gesture Recognition datasets. Through experiments, they have shown that their method shows better performance than baseline methods and their method is not parameter-sensitive, is stable and provides interpretable adaptation results.

I think their method is interesting and motivation is important. However, their experimental results are not convincing enough.
First, they did not compare their method with recent state-of-the-art methods.  For example, there are classifier's discrepancy based adversarial learning method, Saito, Kuniaki, et al. "Maximum classifier discrepancy for unsupervised domain adaptation.". In addition, they did not compare with "A dirt-t approach to unsupervised domain
adaptation", which they cited in the paper. I think their method is for stable and interpretable adversarial learning. So, it does not have to outperform other methods in accuracy. However, they need to show some superiority over these representative adversarial methods.
Second, their experiment is only on digits and WIFI datasets. Is the method effective for object recognition datasets, such as Office or OfficeHome? This is an important question to be addressed because the two datasets are benchmark domain adaptation dataset and the behavior on this dataset will show how this method is applicable to various datasets. I would say that the method does not have to outperform state-of-the art methods for these datasets, but they need to show how the method works on this dataset with respect to stability and interpretability.

In addition, this method seems to have clear connection with "Zhao, Junbo, Michael Mathieu, and Yann LeCun. "Energy-based generative adversarial network." arXiv preprint arXiv:1609.03126 (2016)". They need to add this paper to a reference  and explain some connections.

To sum up, due to the two questions listed above, I think this paper is marginally below the acceptance threshold. Please respond to the questions.


**Experience Assessment:**

I have published in this field for several years.

**Review Assessment: Checking Correctness Of Derivations And Theory:**

I carefully checked the derivations and theory.

**Review Assessment: Checking Correctness Of Experiments:**

I carefully checked the experiments.

**Review Assessment: Thoroughness In Paper Reading:**

I read the paper at least twice and used my best judgement in assessing the paper.

---

> ### Author Response · Authors · 2019-11-14
> **We added the experiments on large-scale dataset and compare our method to the representative methods.**
>
> Thanks for your valuable comments. We agree with your comments and add an experiment as well as more comparisons accordingly.
>
> - Compare their method with recent state-of-the-art methods (such as MCD, DIRT-T, etc).
> Answer: We agree that we should compare our method with some recent state-of-the-art adversarial methods. We compare our method with GTA that employ a generative model to generate source-like features, CDAN that preserves more discriminability in DAT, and MCD that employs the classifier discrepancy. As shown below, we list the comparisons on 3 digit adaptation tasks, where our method outperforms them and MDAT is more computational-efficient.
>
> Method      M-U      U-M       S-M
> GTA             97.8      98.2       91.7
> CDAN          93.9      96.9       88.5
> MCD            96.5      94.1       96.2
> MDAT          98.6      98.4       97.4
>
> As to DIRT-T, it consists of the adversarial training (VADA) and the boundary adjustment module. As the boundary adjustment is an extra technique, we compare MDAT with VADA for fair. The MDAT could be further improved by adding the techniques such as DIRT-T. Note that they employ a 9 Conv layer network while we only use 3 Conv layer as same as DANN and MCD. We compare MDAT with VADA based on these two models.
>
> Method       M-U      U-M       S-M
> VADA-3        96.4      95.5       83.5
> VADA            98.5      97.3       97.9
> MDAT           98.6      98.4       97.4
>
> - Is the method effective for object recognition datasets, such as Office or OfficeHome?
> Answer: We totally agree with it to enrich the availability of our approach on more challenging task and applications. As Office-31 is very small and imbalanced, we add the experiment on the more challenging benchmark, i.e. Office-Home. We compare it with the novel domain alignment approach including DAN, DANN, JAN and CDAN. The proposed ARN with MDAT achieves the best average accuracy of 65.1%.
>
> ------------------------------------------------------------------------------------------------------------------------------------------------
> Method	Ar->Cl	Ar->Pr	Ar->Rw	Cl->Ar	Cl->Pr	Cl->Rw	Pr->Ar	Pr->Cl	Pr->Rw	Rw->Ar	Rw->Cl	Rw->Pr Avg
> ResNet-50	34.9	50.0	58.0	37.4	41.9	46.2	38.5	31.2	60.4	53.9	41.2	59.9	46.1
> DAN	43.6	57.0	67.9	45.8	56.5	60.4	44.0	43.6	67.7	63.1	51.5	74.3	56.3
> DANN	45.6	59.3	70.1	47.0	58.5	60.9	46.1	43.7	68.5	63.2	51.8	76.8	57.6
> JAN	45.9	61.2	68.9	50.4	59.7	61.0	45.8	43.4	70.3	63.9	52.3	76.8	58.3
> CDAN	49.0	69.3	74.5	54.4	66.0	68.4	55.6	48.3	75.9	68.4	55.4	80.5	63.8
> MDAT	51.3	69.7	76.2	59.5	68.3	70.0	57.2	48.9	75.8	69.1	54.3	80.6	65.1
> ------------------------------------------------------------------------------------------------------------------------------------------------
>
> - In addition, this method seems to have clear connection with "Zhao, Junbo, Michael Mathieu, and Yann LeCun. "Energy-based generative adversarial network."
> Answer: We will supplement the references. Thanks.
>
> Reference
> - Long, Mingsheng, et al. "Conditional adversarial domain adaptation." Advances in Neural Information Processing Systems. 2018.
> - Saito, Kuniaki, et al. "Maximum classifier discrepancy for unsupervised domain adaptation." Proceedings of the IEEE Conference on Computer Vision and Pattern Recognition. 2018.
> - Sankaranarayanan, Swami, et al. "Generate to adapt: Aligning domains using generative adversarial networks." Proceedings of the IEEE Conference on Computer Vision and Pattern Recognition. 2018.
> - Zhao, Junbo, Michael Mathieu, and Yann LeCun. "Energy-based generative adversarial network." arXiv preprint arXiv:1609.03126 (2016).
> - Shu, Rui, et al. "A dirt-t approach to unsupervised domain adaptation." arXiv preprint arXiv:1802.08735 (2018).

---

### Official Review · AnonReviewer1 · 2019-10-23
**Official Blind Review #1**

**Rating:** 6

**Review:**

This work proposes Adversarial Reconstruction Network (ARN), a network architecture, and Max-margin Domain-Adversarial Training (MDAT), an objective and training procedure for unsupervised domain adaptation. Similar to domain adversarial approaches, the generator aims at finding domain invariant representation while the discriminator now monitors the reconstruction loss of the source and target data using hinge-like lose. The method is very similar to some of the existing works in the literature. Experiment results on the standard digit datasets and the WiFi gesture recognition dataset show that the proposed method outperforms other alternatives.

Pros:
- The writing is good
- Satisfactory empirical results

Cons:
- The proposed method is very similar to certain methods in the literature

Detail comments:
(1) The proposed loss function Eq.(8) is very similar to the contrastive loss proposed by Hadsell et al. (2006, Eq.(4)), which is used in Siamese GAN variants (Juefei-Xu et al. 2018, Hsu et al. 2019). Thus essentially the proposed method is an application of an existing GAN technique. Its novelty is limited.

(2) Experiments
- How are the hyperparameters selected? It is essential to specify the selection criteria when labeled target data is not available.
- What does the * in DRCN* mean in Table 1?
- ARN w.o. MDAT may not be the best alternative since the target data is ignored in the reconstruction and the discriminator is not discriminating anymore. A more reasonable alternative would be to ignore the margin and minimize L_r(x^s)-L_r(x^t) to see the effect of the margin.

(3) In Eq.(2), y is said to be the predicted domain label (-1 or +1), which could be not accurate according to the common hinge loss definition.

Typos:
- In Eq.(1), there is a missing D in the first term. D_s should be \mathcal{D}_s to match previous notation.
- In Eq.(2), the "0," is not meaningful given the definition of []^+.

Refs
- Hadsell, R., Chopra, S. and LeCun, Y., 2006, June. Dimensionality reduction by learning an invariant mapping. In 2006 IEEE Computer Society Conference on Computer Vision and Pattern Recognition (CVPR'06) (Vol. 2, pp. 1735-1742). IEEE.
- Juefei-Xu, F., Dey, R., Boddeti, V.N. and Savvides, M., 2018. RankGAN: A Maximum Margin Ranking GAN for Generating Faces. In Asian Conference on Computer Vision (pp. 3-18). Springer, Cham.
- Hsu, C.C., Lin, C.W., Su, W.T. and Cheung, G., 2019. SiGAN: Siamese generative adversarial network for identity-preserving face hallucination. IEEE Transactions on Image Processing, 28(12), pp.6225-6236.

# Update after rebuttal

Thank you for the response and additional experiment results. I agree that MDAT and SiGAN are not using the contrastive loss in the same way, but claiming that they are "totally different" can be misleading and overstated. It would be better if the paper includes proper discussion about the contrastive loss from the literature and distinguish the particularities between MDAT and SiGAN. Overall, I think the proposed method shows some prosperity thus I have increased my score accordingly.

**Experience Assessment:**

I have published one or two papers in this area.

**Review Assessment: Checking Correctness Of Derivations And Theory:**

I assessed the sensibility of the derivations and theory.

**Review Assessment: Checking Correctness Of Experiments:**

I assessed the sensibility of the experiments.

**Review Assessment: Thoroughness In Paper Reading:**

I read the paper at least twice and used my best judgement in assessing the paper.

---

> ### Author Response · Authors · 2019-11-14
> **SiGAN uses recontruction term in their paper but it is totally different from ours.**
>
> Thanks for your constructive comments. We think some of the contents in the paper are misunderstanding, and we hope we can address them as below.
>
> (1) Is it similar to SiGAN?
> Answer: No. The forms and the objectives of the loss in two papers are totally different. SiGAN applies a contrastive loss to the reconstruction of the two images, which aims to draw two similar images closer. In MDAT, we propose a margin loss for the reconstruction of the images from the source domain and the target domain. Our objective is to leverage the reconstruction network as a domain classifier that can transfer richer information (i.e. pixel-level domain alignment directly in adversarial training) and have better convergence (i.e. more effective gradients), compared with the classifier binary domain classifier, while SiGAN still leverage the classic binary domain discriminator in DCGAN. SiGAN only regards the contrastive loss as a regularizer during adversarial training, but ours is a new adversarial scheme based on reconstruction. Also, our loss function is more computational-efficient compared to SiGAN.
>
> (2) Experiments
> - How are the hyperparameters selected?
> Answer: We obtain the hyperparameters by cross-validation on the small data SVHN$to$MNIST, and then apply the same hyperparameters for all the experiments. Note that one important novelty is that MDAT is not sensitive to the hyperparameter. As shown in Table 3, as long as the margin m>=1.0 & alpha<=0.1, the results are very robust. Moreover, compared to DANN, the MDAT can always converge with the effective gradients by the reconstruction network.
>
> - What does the * in DRCN* mean in Table 1?
> Answer: Sorry, we actually miss some notes here. This means that we reproduce DRCN to fill some results as they do not provide the result for SYN->SVHN.
>
> - ARN w.o. MDAT may not be the best alternative since the target data is ignored in the reconstruction and the discriminator is not discriminating anymore. A more reasonable alternative would be to ignore the margin and minimize L_r(x^s)-L_r(x^t) to see the effect of the margin.
> Answer: Yes. We conduct ARN w.o. MDAT to verify the efficacy of MDAT in ARN, as an ablation study. It is a good suggestion to just ignore the margin. However, in Table 3, we have shown that if the margin is small (m<0.5), the results are very similar to the standard DAT (DANN),which means that when a small margin or no margin is applied, MDAT has the similar mechanism as DANN. We further added the experiments when m=0 as the reviewer suggested, and supplement the results to Table 3, which further prove the effectiveness of the margin.
>
> m       0       0.1   0.3   0.5   0.7   1.0   2.0   5.0   10.0
> acc    64.3  64.5 75.2 90.0 92.6 96.0 97.4 97.7 96.7
>
> -  In Eq.(2), y is said to be the predicted domain label (-1 or +1), which could be not accurate according to the common hinge loss definition.
> Answer: Yes. We agree with it. There is a little difference that we should point out here. Since we use the hinge loss for domain adaptation, we define the domain label (-1 and +1) in advance that directly confroms with the illustration of the main part (Eq.8). The intuition is the same as the common hinge loss that tell two categories apart from a margin. We will add the explanation after Eq.(2).
>
>
> -Typos.
> We have revised the typos according to the comments.
>
> Refs
> - Hsu, C.C., Lin, C.W., Su, W.T. and Cheung, G., 2019. SiGAN: Siamese generative adversarial network for identity-preserving face hallucination. IEEE Transactions on Image Processing, 28(12), pp.6225-6236.
> - Ghifary, Muhammad, et al. "Deep reconstruction-classification networks for unsupervised domain adaptation." European Conference on Computer Vision. Springer, Cham, 2016.

---

### Official Review · AnonReviewer4 · 2019-11-02
**Official Blind Review #4**

**Rating:** 3

**Review:**


###Summary###
This paper proposes Max-margin domain adversarial training (MDAT) to tackle the problem of transferring knowledge from a rich-labeled source domain to an unlabeled target domain. This is achieved by designing an adversarial reconstruction network. The proposed MDAT stabilizes the gradient by replacing the domain classifier with a reconstruction network.

The motivation of the proposed network is based on the observations that the traditional domain-adversarial training is vulnerable in the following aspects:1) the training procedure of the domain discriminator is unstable, 2) it only considers the feature-level alignment, 3) it lacks the interpretable explanation for the learned feature space.

In the proposed method, the Adversarial Reconstruction Network (ARN) consists of a shared feature extractor, a label predictor, and a reconstruction network. The reconstruction network only focuses on reconstructing samples on the source domain and pushing the target domain away from a margin. The feature extractor tries to confuse the decoder by learning to reconstruct samples on the target domain.

The paper performs experiments on several domain adaptation tasks on digit datasets. The experimental results demonstrate the effectiveness of the proposed results over several baselines such as DANN, ADDA, CyCADA, CADA, etc.

The paper also provides empirical analyses such as t-SNE embedding, plotting the loss, etc. to illustrate the effectiveness of the proposed approach.

### Novelty ###

The model proposed in this paper is extended from the domain adversarial training approach. To stabilize the gradient, the model replaces the domain classifier with a reconstruction network. In this way, the discriminator only discriminates the reconstructed data from the source domain. This idea is interesting and provides some novelty.

###Clarity###

Overall, the paper is well organized and logically clear. The claims are well-supported by the experiments. The images are well-presented and well-explained by the captions and the text.

###Pros###

1) The paper proposes a Max-margin based approach to tackle domain adaptation. Instead of leveraging the domain discriminator to discriminate the source from the target, this paper utilizes a reconstructor to push the target domain far away from the margin. The idea is interesting and heuristic to the domain adaptation research community.
2) The experimental results on digit benchmark demonstrate the effectiveness of the proposed method over other baselines including the most state-of-the-art ones.

3) The paper provides many analyses to demonstrate the effectiveness of the proposed method.

###Cons###


1) The experimental part of this paper is weak. The paper only provides experimental results on the digit recognition experiments, which is not enough to demonstrate the effectiveness and robustness of the proposed approach. Further experimental results on image recognition or NLP task is desired.

It will be also interesting to see how does the proposed method perform on large-scale datasets such as DomainNet and Office-Home dataset:
DomainNet: Moment Matching for Multi-Source Domain Adaptation, ICCV 2019. http://ai.bu.edu/DomainNet/
Office-Home: Deep Hashing Network for Unsupervised Domain Adaptation, CVPR 2017. http://hemanthdv.org/OfficeHome-Dataset/
2) The organization and presentation of this paper should be polished.

Based on the summary, cons, and pros, the current rating I am giving now is weak reject. I would like to discuss the final rating with other reviewers, ACs.



**Experience Assessment:**

I have published one or two papers in this area.

**Review Assessment: Checking Correctness Of Derivations And Theory:**

I assessed the sensibility of the derivations and theory.

**Review Assessment: Checking Correctness Of Experiments:**

I carefully checked the experiments.

**Review Assessment: Thoroughness In Paper Reading:**

I read the paper at least twice and used my best judgement in assessing the paper.

---

> ### Author Response · Authors · 2019-11-14
> **We supplement the experiment on Office-Home.**
>
> Thanks for the comments. We realize that it is necessary to conduct the experiments on large-scale image datasets. As the reconstruction part aims to tackle the pixel-level domain shift, such shift usually exists in image or image-like dataset such as our WiFi recognition dataset.
>
> - The experimental part of this paper is weak.
> Answer: We have added the experiment on Office-Home. The experimental results are shown below. In comparison, the proposed approach achieves 65.1% mean accuracy, outperforming other domain alignment approaches. MDAT shows significant improvement against the standard DAT (DANN) by 7.5%.
> ------------------------------------------------------------------------------------------------------------------------------------------------
> Method	Ar->Cl	Ar->Pr	Ar->Rw	Cl->Ar	Cl->Pr	Cl->Rw	Pr->Ar	Pr->Cl	Pr->Rw	Rw->Ar	Rw->Cl	Rw->Pr Avg
> ResNet-50	34.9	50.0	58.0	37.4	41.9	46.2	38.5	31.2	60.4	53.9	41.2	59.9	46.1
> DAN	43.6	57.0	67.9	45.8	56.5	60.4	44.0	43.6	67.7	63.1	51.5	74.3	56.3
> DANN	45.6	59.3	70.1	47.0	58.5	60.9	46.1	43.7	68.5	63.2	51.8	76.8	57.6
> JAN	45.9	61.2	68.9	50.4	59.7	61.0	45.8	43.4	70.3	63.9	52.3	76.8	58.3
> CDAN	49.0	69.3	74.5	54.4	66.0	68.4	55.6	48.3	75.9	68.4	55.4	80.5	63.8
> MDAT	51.3	69.7	76.2	59.5	68.3	70.0	57.2	48.9	75.8	69.1	54.3	80.6	65.1
> ------------------------------------------------------------------------------------------------------------------------------------------------
>
> - The organization and presentation of this paper should be polished.
> Answer: We will polish the presentations carefully.
>
> Reference
> - Venkateswara, Hemanth, et al. "Deep hashing network for unsupervised domain adaptation." Proceedings of the IEEE Conference on Computer Vision and Pattern Recognition. 2017.
> - Long, Mingsheng, et al. "Conditional adversarial domain adaptation." Advances in Neural Information Processing Systems. 2018.

---

### Decision · Program_Chairs · 2019-12-19

**Decision:**

Reject

**Comment:**

This paper proposes max-margin domain adversarial training with an adversarial reconstruction network that stabilizes the gradient by replacing the domain classifier.

Reviewers and AC think that the method is interesting and motivation is reasonable. Concerns were raised regarding weak experimental results in the diversity of datasets and the comparison to state-of-the-art methods. The paper needs to show how the method works with respect to stability and interpretability. The paper should also clearly relate the contrastive loss for reconstruction to previous work, given that both the loss and the reconstruction idea have been extensively explored for DA. Finally, the theoretical analysis is shallow and the gap between the theory and the algorithm needs to be closed.

Overall this is a borderline paper. Considering the bar of ICLR and limited quota, I recommend rejection.